# Comparison between Intramuscular and Intranasal Administration of Sedative Drugs Used for Piglet Castration

**DOI:** 10.3390/ani14162325

**Published:** 2024-08-12

**Authors:** Andreas Breitenlechner, Moritz Bünger, Ursula Katharina Ruczizka, Marlies Dolezal, Ulrike Auer, Andrea Buzanich-Ladinig

**Affiliations:** 1Clinical Unit of Anaesthesiology and Perioperative Intensive-Care Medicine, University of Veterinary Medicine Vienna, 1210 Vienna, Austria; ulrike.auer@vetmeduni.ac.at; 2Clinical Centre for Population Medicine in Fish, Pig and Poultry, Clinical Department for Farm Animals and Food System Science, University of Veterinary Medicine Vienna, 1210 Vienna, Austria; moritz.buenger@vetmeduni.ac.at (M.B.); ruczizka@tg-oe.at (U.K.R.); andrea.ladinig@vetmeduni.ac.at (A.B.-L.); 3Platform for Bioinformatics and Biostatistics, University of Veterinary Medicine Vienna, 1210 Vienna, Austria; marlies.dolezal@vetmeduni.ac.at

**Keywords:** piglet, castration, intranasal, intramuscular, azaperone, alfaxalone, midazolam, lidocaine

## Abstract

**Simple Summary:**

Castration of male piglets without anaesthesia is facing increasing rejection in society and is also questionable for animal welfare reasons. In this study, the intranasal application of azaperone, azaperone combined with alfaxalone and azaperone combined with midazolam in various doses was tested for its suitability to ensure adequate sedation in combination with intratesticular local anaesthesia during the surgical castration of male piglets. Compared to intramuscular application, intranasal application consistently showed a poorer quality of sedation. Therefore, the intranasal application method cannot be recommended.

**Abstract:**

The aim of this study was to test the intranasal administration of different anaesthetics in piglets less than seven days of age undergoing castration for their suitability for providing good-quality sedation and short induction and recovery time with minimal stress. Azaperone alone at a high (5 mg/kg), medium (3 mg/kg) and low dosage (2 mg/kg) and in two combinations with either alfaxalone or midazolam were applied intramuscularly (i.m.) or intranasally (i.n.) to 120 healthy piglets. Compared to intramuscular application, intranasal application showed longer induction times, shorter recovery times and higher scores for defence and vocalisation. In conclusion, the intranasal protocols did not meet the requirements in all groups and their use can therefore not be recommended. A rapid induction phase and good quality of sedation could not be guaranteed.

## 1. Introduction

The surgical castration of male piglets is common practice in most pig-producing countries. The main purpose of castration is to avoid unwanted odour (“boar taint”), which occurs in 10–75% of intact boars from the age of about five months and is mainly caused by androstenone and skatole [1]. According to the current legal situation in Austria, the castration of male piglets less than seven days old is allowed without anaesthesia, provided that post-operative analgesia is administered [2]. However, the castration of piglets without anaesthesia is increasingly rejected by society and is also questionable from an animal welfare point of view, as castration causes considerable pain during the procedure, but also significant behavioural and pain-related changes for up to four days afterwards [1,3,4,5]. The development of an appropriate pain/anaesthesia management protocol for intra- and post-operative pain would improve animal welfare in commercial pig production. A good sedation or anaesthesia for piglet castration must be able to fulfil several requirements. These include effectiveness, i.e., the anaesthesia must provide sufficient pain relief to make the procedure as stress-free as possible for the piglet. This can be assessed based on no or minimal reaction to handling and the surgical procedure. Criteria such as vocalisation and defensive movements are used for this purpose. The chemical restraint method should be practicable and easy to integrate into the piglet castration procedure. The protocol should be safe and easy to administer and provide analgesia with rapid onset and recovery [6]. The chemical restraint should remain effective throughout the procedure to ensure adequate pain relief. Chemical restraint should not cause undesirable side effects or complications and the costs should be reasonable and proportionate to the benefits. The fulfilment of these requirements is crucial to determine whether piglet castration can be performed with a particular anaesthesia protocol that includes the route of administration. Compared with intravenous and intramuscular techniques, the intranasal administration of sedatives is both non-invasive and painless [7]. Direct transport from the nose to the brain via the olfactory and trigeminal nerve pathways bypasses the blood–brain barrier, resulting in rapid absorption [8]. Its ease of use has the advantage that it could be carried out by farmers themselves, thereby reducing costs. Alfaxalone and midazolam are currently not authorised for pigs in the European Union [9]. Their use is therefore not possible at the moment, and one aim of this study was therefore to test their basic suitability for intranasal sedation. In Austria, azaperone may be supplied to specially trained farmers, but only for the indication of tranquilising aggressive sows. Sedation or anaesthesia for castration carried out by farmers is not permitted in Austria and must be performed by a veterinarian [10,11]. The aim of this study was to find out whether the intranasal administration of different anaesthetics and combinations of anaesthetics in piglets undergoing castration at less than seven days of age can achieve a good quality of sedation with an average sedation score <1 (see below) and a similar short induction and recovery time compared to intramuscular administration. We tested the hypothesis that the intranasal application of the drugs used can achieve equivalent sedation to intramuscular application.

## 2. Materials and Methods

A total of 120 healthy male piglets (domestic pig, species *Sus scrofa domestica*, Large White × Pietrain), aged four to seven days, were used. All piglets underwent a clinical examination before the start of the experiment and animals that were sick or showed deviations from normal anatomical condition were not included in this study. The study protocol was approved by the institutional ethics and animal welfare committee and the national authority according to §§ 26ff. of the Animal Experiments Act, Animal Experiments Act 2012 (approval number 2020-0.148.805).

### 2.1. Study Design

A randomised and blinded experimental study design was used. The experiment was split over two days. On the days of the experiment, the piglets were weighed and received an ear tag and were subsequently fixed in the nest for 25 min to allow them to calm down. The piglets were divided into twelve treatment groups of ten animals. The allocation into treatment groups started with assigning the first piglet of the study day to group 1, then the second piglet to group 2, until group 12. Then, the cycle of allocation started in the same rhythm again. Each time, six piglets were sedated consecutively at three-minute intervals to be able to work more time-efficiently and thus create more practical conditions. Sedation was carried out by a single veterinarian throughout the course of the trial; the two other veterinarians, who carried out castration and assessment, were blinded and had no knowledge which piglet was sedated by which protocol. The drugs were applied either intranasally using an intranasal mucosal atomisation device (MAD NadalTM, Teleflex Medical, Morrisville, NC, USA), or by intramuscular injection into the neck muscles using a 23G needle (Sterican 23G × 1”, B.Braun Petzold GmbH, Melsungen, Germany). After application, the piglets were put back into the heated nest. After approximately ten minutes, 0.2–0.3 mL lidocaine (Xylanaest^®^ 2% with epinephrine 1:200,000, Gebro Pharma GmbH, Fieberbrunn, Austria) was applied into each testicle using a 27G cannula (Sterican 27G × ¾″, B.Braun Petzold GmbH, Germany). This procedure took less than a minute. A further waiting period of 25 min until castration was chosen to allow lidocaine to diffuse from the testis into the spermatic cord. Analgesia for castration is provided by lidocaine, as alfaxalone, azaperone and midazolam do not possess analgesic properties [12]. Castration was performed using scalpels by making two short skin incisions, advancing the testis from the scrotum and depositing the spermatic cord, including blood vessels and muscles lege artis, by using an emasculator. The procedure was performed by a single experienced member of the study group throughout the entire experiment and took less than a minute. Before being returned to the piglet nest, each piglet was injected with 0.4 mg/kg meloxicam (Metacam^®^ 20 mg/mL, Boehringer Ingelheim, Ingelheim am Rhein, Germany) into the neck muscles. As soon as the piglets regained their ability to stand, they were returned to the sow. The piglet pens were video-monitored so that an evaluation of the introduction and recovery times could be carried out retrospectively. The castration process was also recorded on video. All induction and recovery times were assessed retrospectively using the video recordings. The scoring was always carried out in real time. Our requirements for a positive evaluation of a protocol were a good quality of sedation with an average of less than one defensive movement and vocalisation, a short induction time of 10 min maximum and a recovery time of 60 min maximum.

### 2.2. Anaesthesia Protocols

Azaperone alone at a high (5 mg/kg), medium (3 mg/kg) and low dosage (2 mg/kg) and with the same dosages in two combinations with either alfaxalone or midazolam were applied intramuscularly (i.m.) or, when each in the high dose, intranasally (i.n.) (Table 1).

### 2.3. Data Collection

#### 2.3.1. Time

Induction time (mean_induction) was the time from drug application to lateral recumbency. Recovery time (mean_recovery) was the time measured from the end of castration to return to the sow. All times are given in minutes. Not all animals reached lateral recumbency within 35 min. As castration took place after this time, these animals were included in the statistics with the maximum possible 35 min.

#### 2.3.2. Sedation Score

The quality of sedation was assessed using a simple descriptive scale modified from Michou et al. [13], Santos et al. [14] and Vullo et al. [15] (Table 2). This scoring system was chosen for its simplicity of use and sufficiency for the purposes of the study.

Defensive movements and vocalisations were scored during the following:Lifting and fixation before application of local anaesthetic (LA) (mean_scoring_handling_LA);Application of LA (mean_scoring_applicaton_LA);Lifting and fixation before the start of the actual castration (mean_scoring_handling_castration);Castration (mean_scoring_castration).

A total mean score was calculated as the mean of the four previous scores (mean_total_scoring).

### 2.4. Statistics

All statistical analyses were performed in R v4.0.2 [16]. Individual scores or durations were aggregated to groups of phenotypes, each describing a different aspect of the procedure. Arithmetic means as aggregate summary statistics for time points (mean_induction, mean_recovery) and scores (mean_scoring_handling_LA, mean_scoring_applicaton_LA, mean_scoring_handling_castration and mean_scoring_castration, mean_total) were calculated. Means for these phenotype groups were then log10-transformed after adding a constant of one to each observation and used as a univariate response in a linear mixed model applying the lmer function in the lme4 v1.1-27 package [17]. A combined fixed categorical main effect of drug treatment, dosage and application was fitted. Drug treatment and dosage each had three levels, Azaperon (Aza), Azaperon/Alfaxalon (Aza/Alf) and Azaperon/Midazolam (Aza/Mid), low-medium-high (l-m-h) and each application two levels, intramuscular and intranasal (i.m. and i.n.), respectively. Note that for ethical reasons, a fully crossed design could not be performed, i.e., no low or medium dosage levels for the intranasal application, which would have allowed for fitting drug, dosage and application as separate main effects, a three-way interaction and the lower-level two-way interactions between them. Nevertheless, the combined fixed main effect of drug–dosage–application still enabled testing for differences between dosage–application levels within the drug and for drug differences within dosage–application levels. This resulted in four dosage–application levels (h_i.m., h_i.n., l_i.m. and m_i.m.) for each of the three drug treatments (twelve factor levels overall with measures for ten male piglets for each). Animal identification of the mother sow was fitted as a random intercept to account for a possible covariance structure as multiple piglets from the same litter were included in our experiments. To achieve optimal estimates for the fixed effect in this model, the REML option was set to false to obtain maximum likelihood estimates. It was checked and validated that the log10-transformed responses met all necessary assumptions for linear mixed models. Residuals and random intercepts were normally distributed and residuals were homoscedastic. First, a full-reduced model comparison was performed using the function anova in base R with the test = ”chisq” option for all log10-transformed aggregated phenotype-group means to test for the overall significance of the combined drug–dosage–application effect. The full and reduced models both included a random intercept effect of the mother sow. A significant *p*-value for such a hierarchical model comparison means that at least one level of the combined fixed drug–dosage–application effect is significantly different from at least one other level. *p*-values were adjusted for multiple testing with a Benjamini–Hochberg correction [18]. Subsequently, estimated marginal means for each factor level were calculated and it was tested for differences between drugs within dosage–application levels and for dosage–application differences within the drug treatment with the emmeans v1.6.1 package [19]. The results of our hypothesis testing were visualised via bar plots on back-transformed estimated marginal means, which define the height of the bar, with the ggplot2 v3.3.3 package [20]. The fitted model is shown as a black dot-and-whisker plot representing upper and lower 95% confidence intervals. *p*-value brackets for contrasts of interest between estimated marginal means were created with the ggpubr v0.4.0 package [21]. The *p*-values shown were corrected for multiple testing across all contrasts of interest across all response variables but separately for each research question, i.e., separately for drug differences within dosage–application levels [four dosage–application levels for each of the three drug treatments multiplied by nine response variables: a total multiple testing load of one hundred and eight], and dosage–application differences within drug treatment [eighteen contrasts multiplied by nine response variables: a total multiple testing load of one hundred and sixty-two], using the false discovery rate approach proposed by Benjamini and Hochberg [18]. Significance was declared at a multiple testing corrected for 5% false discovery rate. A colourblind-friendly palette was created with the RColorBrewer v1.1-2 package [22]. Principal component analysis was performed with the factoextra v1.0.7 package [23]. Data were prepared using functions from the tidyverse v1.3.1 [24], dplyr v1.0.6 [25], tidyselect v1.1.1 [26] and stringr v1.4.0 packages [27].

## 3. Results

The piglets had a mean age of 5 ± 1 (4–7) days and a mean weight of 2.45 ± 0.55 (1.35–4.20) kg and were not different between study groups.

### 3.1. Induction Time

The mean induction time ranged from 3.82 min in G10 to 18.19 min in G4 (Table 3). Azaperone alone (G1, G2, G3, G4) had the longest induction time regardless of the route of application (Appendix A). The induction times in G8 and G12, the intranasal groups, were significantly longer (*p* < 0.05) than in the respective high (G7, G11) and medium (G6, G10) i.m. groups (Appendix A, Table 3). In G10, all piglets reached a lateral position within 10 min; in G11, 90%; in G6 and G7, 80%; in G2, G5 and G8, 70%; in G3, G9 and G12, 60%; in G1, 40%; and in G4, only 30%. One piglet each in G1, G4 and G5 did not reach lateral recumbency within 35 min.

### 3.2. Sedation Score: Handling for Application of Local Anaesthetics

During lifting and fixation prior to local anaesthetic application, the intranasal groups G4 and G12 showed higher scores than the intramuscular groups. Group G5 showed the highest score. There was an indirectly proportional relationship between dose and score in all intramuscular groups (G1–3, G5–7, G9–11). Higher doses were associated with lower scores and inversely (Appendix A, Table 3). Only G3, G6–7 and G9–11 showed a mean score of <1, which means that in all other groups, most piglets reacted at least once.

### 3.3. Sedation Score: Application of Local Anaesthetics

During the application of the local anaesthetic, G4, G8 and G12 showed higher scores than all the intramuscular groups (Table 3). High Aza/Alf (G7) and Aza/Mid (G11) doses had lower scores than medium intramuscular doses (G6, G10) (Appendix A, Table 3). Only G2–3, G6–7 and G10 showed mean scores of <1, which means that in all other groups, most piglets reacted at least once.

### 3.4. Sedation Score: Handling for Castration

During lifting and fixation prior to castration, piglets in G4, G8 and G12 showed the highest scores. However, the lowest score was seen in the mid-dose group G2 (Appendix A, Table 3). Only G6–7 and G9–11 showed mean scores of <1, which means that in all other groups, most piglets reacted at least once.

### 3.5. Sedation Score: Castration

For castration, intranasal groups G4 and G8 and intramuscular group G5 had the highest scores (Table 3). There were no significantly different scores (*p* > 0.05) between the intranasal and intramuscular groups G1 to G4 and G5 to G8. The high intramuscular doses of Aza/Alf (G7) and Aza/Mid (G11) showed the lowest scores (Appendix A, Table 3). G2, G7 and G9–11 showed mean scores of <1, which means that in all other groups, most of the piglets reacted at least once.

### 3.6. Total Score of All Assessed Steps

Assessing all steps together, the intranasal groups showed the highest total scores, ranging from 1.51 (G8) to 2.01 (G4), and they did not significantly differ from each other (Table 3). The lowest score was achieved by the high intramuscular dose of Aza/Alf (G7) (Appendix A, Table 3). G3, G6–7 and G9–11 showed mean scores of <1, which means that in all other groups, most of the piglets reacted at least once.

### 3.7. Recovery Time

The recovery time in the intranasal groups ranged from 6.80 min (G12) to 8.48 min (G8), respectively. In the intramuscular groups, recovery times ranged from 6.70 min (G5) to 160 min in G11. A recovery time of more than one hour was seen for G7 and G9–G11, which was significantly longer (*p* < 0.05) than for G1-G3 and G5-G6 (Appendix A, Table 3).

## 4. Discussion

In general, the aim to find an intranasal protocol which fulfils the requirements, such as good quality of sedation, short induction and recovery time during piglet castration, could not be achieved. None of the three intranasal groups reached an induction time of less than 10 min. Most of the intramuscular drug combinations also failed to meet the criteria for good and acceptable sedation for castration. However, the scores for handling were significantly higher than those for castration, as local anaesthetics do not affect handling stress [28]. The majority of piglets showed at least one defensive movement and/or vocalisation during castration, at least during the pulling of the spermatic cord (Appendix A, Table 1). Taylor and Weary have shown that the pulling and cutting of the spermatic cord is the most painful part of the castration procedure [29]. Nociceptive responses elicited during castration despite local anaesthesia may be explained by the fact that lidocaine does not readily diffuse through the tunica vaginalis to the cremaster muscle, which is also transacted during the procedure [30]. Although previous studies have shown that local anaesthetics can reduce pain during castration, intraoperative pain is not completely eliminated [31]. Ranheim found that, autoradiographically, the highest concentration of lidocaine is present in the spermatic cord three minutes after intratesticular application [32]. In order to simulate practical conditions, the lidocaine was given 25 min to take effect in this study. The requirements of a score ≤1 at handling and castration were only met by intramuscular azaperone/midazolam (G9–11) and the high intramuscular dose of azaperone/alfaxalone (G7). This suggests that a combination with alfaxalone or midazolam improves the sedative effect compared to azaperone alone. Long recovery times are undesirable as piglets nurse approximately once per hour [33]. Recovery from anaesthesia is a stressful experience, especially for very young piglets. Piglets under anaesthesia must be warmed and possibly dried during the recovery period to avoid life-threatening hypothermia. As young and small piglets in particular do not yet have the ability for sufficient temperature regulation and lack the necessary energy reserves, temperature must be monitored during the recovery phase [34]. A heated flooring system may be better suited to ensure adequate heat supply than a heat lamp, as it warms all piglets equally. However, there is also a risk of hyperthermia if the floor temperature is too high [35]. Older piglets were found to not only regain consciousness faster but were also more coordinated and fitter after a shorter period of time. To reduce the risk of piglets being crushed or trampled, it is suggested to avoid anaesthesia of piglets younger than one week and to separate piglets into groups of similar size and weight [34]. Axiak has shown that the long recovery period when using climazolam in 4–7-day-old piglets can be shortened with sarmazenil [36]. Other studies in older pigs have shown that antagonism with flumazenil can reduce the recovery time when midazolam or zolazepam is used [36,37]. This suggests that flumazenil may have the same effect in younger piglets, although further studies are needed. The high cost of flumazenil and the additional handling of piglets may make its use difficult in practice. In addition, flumazenil has a relatively short half-life and moderate-to-profound resedation may occur if the antagonist dose or duration of anaesthesia is inadequate; so, multiple applications of flumazenil may be required [38]. From the evaluation of the selected protocols, it can be concluded that the intranasal administration of anaesthetics has no advantage over intramuscular administration and can therefore be ruled out for use by farmers. Due to differences in sedation protocols, a direct comparison with our study is difficult, but other studies also consistently showed poorer sedation results with intranasal administration compared to intramuscular application [36,39,40,41]. This supports our conclusion that this form of administration is not recommendable for piglets. The combination of azaperone with alfaxalone or midazolam is also not the ideal solution. In order to achieve adequate sedation with low resistance during castration, higher doses are required, at the expense of a much too long recovery time. Intranasal administration of the drugs presented some difficulties. The piglets showed massive defensive movements during the administration and had to be well fixed accordingly. The nasal application probably caused discomfort as some of the substances were sneezed out. Several factors—such as the viscosity, rheological properties and surface tension of the drug, the actuation force of the sprayer used and the pump design—can influence the characteristics of nasal aerosol generation [42]. The nebulisers were derived from human medicine and were not specifically designed for the noses of the piglets. Future studies should compare whether a custom-made nebuliser and drugs specifically designed for intranasal use would make a difference. Given the massive resistance of piglets to intranasal application, it is questionable whether this method really is less stressful than intramuscular injection. No data are currently available in this regard, so this would be an interesting topic for future studies. The induction time for intranasal application was longer than for intramuscular application in all drug groups and was statistically significant in some cases (Appendix A). We suggest that the strong defence of the piglets during application could be a possible cause of the longer induction times and the higher scores in the intranasal groups. As the substances were partially sneezed out in some cases, it could not be guaranteed that the full dose was absorbed. In the azaperone–alfaxalone groups, the concentration of alfaxalone also caused problems. Alfaxalone is currently only available in a concentration of 10 mg/mL. This resulted in high volumes, which in most cases led to the piglets coughing up or swallowing part of the applied volume. In a pilot study, the ideal volume for intranasal application in piglets of this age group was found to be approximately 0.4 mL on average. For example, in a 2 kg piglet, the high dose of 5 mg/kg would require a volume of 1 mL of alfaxalone alone at the current concentration. Administering large amounts too quickly may result in swallowing of the drug and subsequent delayed absorption from the gastrointestinal tract [43]. A recent 2021 study showed the same problems after intranasal sedation with alfaxalone. Also, in this study, intranasal application of alfaxalone did not provide sufficient sedation to allow for handling, the smooth performance of a non-invasive clinical procedure or the reduction in stress [39]. However, this study was carried out on adult pigs of a different breed, which could possibly lead to deviations. The sedation protocol also consisted of alfaxalone only, which makes a direct comparison with our study difficult, as here, alfaxalone was combined with azaperone. Previous studies have described muscle twitching in pigs after induction with intravenous alfaxalone [44,45,46]. This was not observed in this study, which could mean that either the combination with azaperone attenuates this effect, or that this effect does not occur at all with the intranasal and intramuscular routes of administration. Should a formulation with a higher concentration become available in the future, this would be of interest for further studies. Alfaxalone is a neuroactive steroid compound that potentiates the action of the g-aminobutyric acid A receptor and has successfully been used in previous studies both alone and in combination to induce general anaesthesia and muscle relaxation in pigs. It can be administered to swine both intramuscularly and intravenously [13,14,15,44,45,46]. At clinical doses, the analgesic properties of alfaxalone are negligible [12]. To the authors’ knowledge, there were no studies on the intranasal use of alfaxalone in pigs at the start of the experiment. At the time, the drug was not authorised for use in pigs in the European Union [9]. Azaperone is a tranquilizer of the butyrophenone class approved for use in pigs to prevent aggression, stress and fighting [9,47,48,49]. It is a common sedative, often used in combination with other drugs, and has no analgesic effect [36,50,51]. Azaperone appears to have minimal effects on respiration and may inhibit some of the respiratory depressant effects of general anaesthetics. In pigs, a slight decrease in arterial blood pressure was measured after the intramuscular injection of azaperone [12]. Previous studies have shown that a rapid onset of action and pigs treated with azaperone typically respond by lying down within minutes of injection [12,48]. The duration of action is approximately 2–3 h in young pigs and 3–4 h in older individuals [12]. Svoboda et al. [41] found that intranasal application of azaperone requires a dose of 4 mg/kg to achieve the same effect of the standard recommendation of 2 mg/kg for intramuscular application. The results of the study indicate that the duration of sedation is also prolonged with increasing dosage. With 60 min compared to 15 min, the groups with intranasal application of azaperone showed a significantly longer duration until onset of sedation than with intramuscular application. While in our study, the intranasal group also showed a longer induction time compared to the intramuscular groups, the animals showed clear signs of sedation on average after 18.19 min. The nasal cycle in pigs has been described by Campbell and Kern [52] and could be one possible explanation for this discrepancy. Studies have shown that the nasal cycle in humans occurs as a phenomenon in which the nose undergoes regular periods of congestion and decongestion. Nasal mucociliary clearance is reduced during periods of congestion, which in turn may affect drug absorption [52,53]. Axiak et al. [36] attempted intranasal application of ketamine, climazolam and azaperone, although the authors found no existing studies of intranasal application in combination with alfaxalone or midazolam. Midazolam is a benzodiazepine and produces muscle relaxation and sedation with minimal cardiorespiratory effects in pigs [15,54,55]. The exact mechanism of action is not yet known, although antagonism of serotonin, increased release and/or facilitation of gamma-aminobutyric acid (GABA) activity, and decreased release or turnover of acetylcholine in the CNS are suspected [12]. This drug stands out from others in this class due to its rapid onset and short duration of action [56]. Past studies have shown a rapid onset of action within minutes when applied intranasally in pigs [57]. Midazolam is not approved for pigs in the European Union [9]. Only one dosage was used for midazolam. We referred to the study by Lacoste (2000) [57], in which a higher dose did not provide any advantage over the 0.2 mg/kg used there. As the piglets in said study were only drowsy and still able to walk, we were concerned that lower doses would have an insufficient effect and rather wanted to find out whether the quality/depth of sedation could be improved by adding azaperone at different doses.

Lidocaine hydrochloride can be both injected and administered as a topical agent. Lidocaine inhibits the transmission of stimuli to the brain by preventing the nerve from initiating an action potential [58]. Compared to procaine—which has been evaluated by the European Medicines Agency (EMEA) as a local anaesthetic that can be used in livestock without a maximum residue limit—lidocaine shows some advantages. The onset of action is faster and its duration is longer, there are fewer side effects and it also shows easier distribution in the tissues [30]. Analgesia normally occurs within five minutes of application and lasts 1–2 h [12]. The addition of a vasoconstrictor prolongs the effect. It also reduces the risk of systemic toxicity by decreasing the rate of systemic absorption [59]. After intratesticular application, lidocaine is rapidly transported into the spermatic cord [32]. Lidocaine is currently not approved for use in pigs in the European Union [9].

Meloxicam is a non-steroidal anti-inflammatory drug (NSAID) of the oxicam class inhibiting prostaglandin synthesis via relatively selective inhibition of cyclooxygenase-2 and provides analgesic, antipyretic and anti-inflammatory properties [60]. It is registered for use in piglets [9]. While the administration of meloxicam is helpful in the treatment of post-operative pain and suffering, this drug cannot manage intra-operative pain [61,62].

## 5. Conclusions

Our hypothesis, that intranasal administration of the drugs used can produce equivalent sedation to intramuscular administration, could not be confirmed. This study has shown that intranasal application of the used drugs produced poor sedation and, therefore, is not feasible for clinical practice. All intranasally sedated pigs showed mild sedation, although this was not sufficient to allow for any major manipulations. Regarding intramuscular application, only the medium dosage of the azaperone/alfaxalone combination (G6) met all our criteria. All other combinations showed either too long induction times, unsatisfactory scores or too long recovery times.

## Figures and Tables

**Table 1 animals-14-02325-t001:** Drugs, dosages and routes of application (i.n. = intranasal, i.m. = intramuscular).

Group	Drug	Dosage	Route of Application	Shortcut
G1	Azaperone	2 mg/kg	i.m.	Aza_l_i.m.
G2	Azaperone	3 mg/kg	i.m.	Aza_m_i.m.
G3	Azaperone	5 mg/kg	i.m.	Aza_h_i.m.
G4	Azaperone	5 mg/kg	i.n.	Aza_h_i.n.
G5	Azaperone/Alfaxalone	2 mg/kg/2 mg/kg	i.m.	Aza/Alf_l_i.m.
G6	Azaperone/Alfaxalone	3 mg/kg/3 mg/kg	i.m.	Aza/Alf_m_i.m.
G7	Azaperone/Alfaxalone	5 mg/kg/5 mg/kg	i.m.	Aza/Alf_h_i.m.
G8	Azaperone/Alfaxalone	5 mg/kg/5 mg/kg	i.n.	Aza/Alf_h_i.n.
G9	Azaperone/Midazolam	2 mg/kg/0.2 mg/kg	i.m.	Aza/Mid_l_i.m.
G10	Azaperone/Midazolam	3 mg/kg/0.2 mg/kg	i.m.	Aza/Mid_m_i.m.
G11	Azaperone/Midazolam	5 mg/kg/0.2 mg/kg	i.m.	Aza/Mid_h_i.m.
G12	Azaperone/Midazolam	5 mg/kg/0.2 mg/kg	i.n.	Aza/Mid_h_i.n.

Azaperone (Stresnil^®^ 40 mg/mL, Elanco, Cuxhaven, Germany); alfaxalone (Alfaxan^®^ Multidose 10 mg/mL, Jurox, Dublin, Irland); and midazolam (Dormicum^®^ 5 mg/1 mL, CHEPLAPHARM Arzneimittel GmbH, Greifswald, Germany).

**Table 2 animals-14-02325-t002:** Scoring sheet.

Score	Defensive Movements	Vocalisation
0	None	None
1	Once	Once
2	Multiple times	Multiple times
3	Continuous	Continuous

**Table 3 animals-14-02325-t003:** Estimated marginal means ± standard error (SE): time of induction and recovery after intranasal/intramuscular application of azaperone; mean scoring of movement and vocalisation: low-dose group i.m. (Aza_l_i.m. 2 mg/kg), medium-dose group i.m. (Aza_m_i.m. 3 mg/kg), high-dose group i.m. (Aza_h_i.m. 5 mg/kg) and high-dose group i.n. (Aza_h_i.n. 5 mg/kg); azaperone + alfaxalone: low-dose group i.m. (Aza/Alf_l_i.m. 2 mg/kg + 2 mg/kg), medium-dose group i.m. (Aza/Alf_m_i.m. 3 mg/kg + 3 mg/kg), high-dose group i.m. (Aza/Alf_h_i.m. 5 mg/kg + 5 mg/kg) and high-dose group i.n. (Aza/Alf_h_i.n. 5 mg/kg + 5 mg/kg); azaperone + midazolam: low-dose group i.m. (Aza/Mid_l_i.m. 2 mg/kg + 0.2 mg/kg), medium-dose group i.m. (Aza/Mid_m_i.m. 3 mg/kg + 0.2 mg/kg), high-dose group i.m. (Aza/Mid_h_i.m. 5 mg/kg + 0.2 mg/kg) and high-dose group i.n. (Aza/Mid_h_i.n. 5 mg/kg + 0.2 mg/kg) with ten piglets in each group. Induction time was measured from time of drug application until lateral recumbency was reached. Recovery time was measured from the time of return to the piglet nest after castration until the return to the mother sow.

Groups	Shortcuts	Induction (min)	Mean Scoring Handling LA	Mean Scoring Application LA	Mean Scoring Handling Castration	Mean Scoring Castration	Mean Total Scoring	Recovery (min)
G1	Aza_l_i.m.	14.86 ± 3.43	1.1 ± 0.29	1.25 ± 0.31	1.54 ± 0.23 *	1.19 ± 0.20	1.36 ± 0.18	9.28 ± 3.22
G2	Aza_m_i.m.	10.58 ± 2.50	1.02 ± 0.28	0.80 ± 0.25 *	1.13 ± 0.19 *	0.89 ± 0.18	1.04 ± 0.15 *	13.80 ± 4.63
G3	Aza_h_i.m.	10.44 ± 2.46	0.27 ± 0.18 *	0.60 ± 0.22 *	1.22 ± 0.20 *	1.16 ± 0.20	0.93 ± 0.14 *	43.95 ± 14.00 *
G4	Aza_h_i.n.	18.19 ± 4.17	1.98 ± 0.42	2.19 ± 0.44	2.41 ± 0.30	1.35 ± 0.22	2.01 ± 0.22	7.60 ± 2.71
G5	Aza/Alf_l_i.m.	8.41 ± 2.03	1.24 ± 0.31	1.53 ± 0.35	1.55 ± 0.23	1.40 ± 0.22	1.47 ± 0.18	6.70 ± 2.41
G6	Aza/Alf_m_i.m.	3.89 ± 1.06 *	0.28 ± 0.18 *	0.54 ± 0.21 *	0.67 ± 0.15 *	1.21 ± 0.21	0.74 ± 0.13 *	27.16 ± 8.81 *
G7	Aza/Alf_h_i.m.	4.14 ± 1.09 *	0.08 ± 0.15 *	0.80 ± 0.24 *	0.15 ± 0.10 *	0.69 ± 0.16	0.40 ± 0.10 *	68.33 ± 21.41 *
G8	Aza/Alf_h_i.n.	10.08 ± 2.36	1.03 ± 0.28	2.11 ± 0.43	1.69 ± 0.24	1.14 ± 0.20	1.51 ± 0.19	8.48 ± 2.93
G9	Aza/Mid_l_i.m.	8.51 ± 2.02	0.48 ± 0.20	1.15 ± 0.30 *	0.55 ± 0.14 *	0.79 ± 0.17 *	0.72 ± 0.13 *	70.04 ± 21.95 *
G10	Aza/Mid_m_i.m.	3.82 ± 1.02 *	0.33 ± 0.18 *	0.62 ± 0.22 *	0.49 ± 0.13 *	0.77 ± 0.16 *	0.58 ± 0.12 *	106.08 ± 33.10 *
G11	Aza/Mid_h_i.m.	4.62 ± 1.19 *	0.15 ± 0.16 *	1.21 ± 0.30	0.22 ± 0.11 *	0.65 ± 0.15 *	0.50 ± 0.11 *	160.16 ± 49.64 *
G12	Aza/Mid_h_i.n.	9.95 ± 2.35	1.28 ± 0.32	2.39 ± 0.47	2.18 ± 0.28	1.64 ± 0.25	1.95 ± 0.22	6.80 ± 2.43

* Significantly different from the intranasal group (*p* < 0.05).

## Data Availability

Raw data are available upon request from the corresponding author.

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
