# Peer review of "Comparison between Intramuscular and Intranasal Administration of Sedative Drugs Used for Piglet Castration"

_animals, 2024, doi:10.3390/ani14162325_

Round 1

Reviewer 1 Report

Comments and Suggestions for Authors

To diminish castration pain, they were sedated intranasally with different anaesthetics and doses together with an intratesticular local anaesthetic. 

Introduction

Regarding the title, introduction (aim, L64) and conclusions, it is not clear if the objective is sedation or anaethesia, i,e, reduction of behaviour or reduction of pain sensation.

M&M

L68: The meaning of clinical health status is not clear?

L76: The experiment is performed in 1 day, since each piglet takes 25 + 25 min and how many piglets were observed by 1 person. At the end the investigators are very tired.

L77-83: The sedation and castration were perpormed by 1 person (a qualified vet?) and which person did the observation (trained?). At the end the procedure is not clearly described. 

L139: Statistics: Is the action of the investigator included i.e. person dependend. May be tyred at the end?

Discussion

L295, 296: Is it anaesthesia or sedation?

L296 Stress was not examined.

What is adequate L298, rapid L299, best L303, goal L305, sufficient LL312, significant L314, acceptable L319. This part of the discussion needs to be reconsidered L294-326. 

L336-347 This part needs to be reconsidered. What is the meaning of requirements and goal?

L357-360 Is this the conclusion?

L434-447 Here are conclusions needed and not a summary , Please reconsider the conclusions. 

Reviewer 2 Report

Comments and Suggestions for Authors

Dear authors,

thank you very much for the interesting manuscript about intranasal application of different sedation protocols in protocols for piglet castration.

It is a hot topic, trying to balance animal welfare and economics, therefore, it might be of great interest to the reader. The study design is simple and attempted to represent clinical conditions, however, the statistical analysis seems overcomplicated, considering the low animal numbers and variables of interest. In addition, the presentation of the study and its results needs major revision to make it comprehensive and clear. A thorough differentiation between sedation or anaesthesia would be helpful, as apparently none of the groups provided consistently anaesthesia. From a professional point, these two terms should be clearly distinguished or the term "chemical restraint" can be used instead. 

The discussion is lacking comparison to existing literature with regard to suitable doses, protocol and administration techniques. A proper discussion for the obtained results and reflection of methods used is inadequate. Important references (eg Svoboda et al 2023, Becker et al 2021) are missing.

I have chosen a major revision, as a substantial rewriting is necessary, before it can be reassessed.

I have added some more comments below, this list is just a selection and not complete.

Introduction:

Legal restriction with regard to drug selection for food producing animals and reference to legally allowed people administering sedatives should be included and mentioned. This is not necessarily worldwide transferable, but at least it gives an indication, that there are strict regulations.

- rephrase aim of the study and hypothesis

- throughout the manuscript terms of "stress" and "ethically acceptable" are used without a reference or details. Both parameters are not quantitative, where never assessed in this study (stress) or are used very biased. Please define exactly what you are referring to and explain how you distinguish between pain and stress.

- I am not sure, whether the process of intranasal application is less stressful than intramuscular - please clarify/comment

M and Ms:

- more details: was the investigator the same in all piglets? how was observation performed (difference between "sedation score" and assessed timings)

- check dosages used - was MIdazolam used always at the same dose, whereas Alfaxalon was altered?

- sedation score: reference to dog paper is unclear, also there are established/published sedation scores - compare to these (pros and cons), also how/why it was established/validated

- very complex. unnecessary complicated statistics, missing power analysis or group size allocation

- recovery time should be based on time from injection until return, rather than time after finishing castration, please also record average time in each group for LA and castration and duration

Results:

- unclear what is represented in the graphs, mainly it shows a repetition of the table

- inconsistent use of group names

- qualitative summary of results (eg all pigs recumbent at some time?), most of the pigs reacted during castration in xyz missing to represent findings from the table

- discussion of existing doses and protocols, compare relevant literature with your results

- does the choosen protocol really reflect a clinical setting (eg 25 min between LA administration and castration?)

- comment on physiological effects of sedation - eg. lateral recumbenct pigs for > 35 min left alone - pleaes comment on potential side effects with regard to safe drug choice

- missing: study type/design

conclusions are not necessarily drawn from results, but biased (consider use of non-inferior or superior hypothesis - either you are comparing 12 different groups and try to find the best or you want to see whether the 3 intranasal protocols are as good as the IM protocols)

why has not an established protocol been included as reference?

Comments on the Quality of English Language

please check decimal indicators - that is not consistent

Round 2

Reviewer 1 Report

Comments and Suggestions for Authors

Author Response

Dear Reviewer 1

Thank you very much for your review. I am pleased that there are no outstanding issues. Your comments have significantly improved the manuscript.

Kind regards
 Andreas Breitenlechner

Reviewer 2 Report

Comments and Suggestions for Authors

Dear authors, thank you very much for revision of the manuscript about different anaesthesia/restraint protocols im pigs.

The quality and data presentation as much improved, however, i still have some minor comments:

Line 49, 52, 53

Use “chemical restraint” instead of anaesthesia, as you are still referring to both – anaesthesia and/or sedation

Line 64: add “similar short induction and recovery time…”

Line 66-72 – describing the background of your study but not the aim/hypothesis – move further up in introduction to an appropriate place

Study goals added, however, your positive outcome is based on the sedation score, which has not introduced at this point yet and is also not mentioned in the study goal. Maybe include in the sentence (“average sedation score < 1, see below”)

The explanation for the choice of midazolam dose is not part of the M and Ms, move to discussion

Data collection

Please clarify, which data was obtained “realtime” and which one “retrospectively by video analysis”. Include this info also in the relevant section and how it was done

Line 218: komma as decimal instead of dot

Results:

Induction times – you mention above (line 138-140), that not all animals reached lateral recumbency within 35 minutes. However, this data is not presented in paragraph 3.1 (unclear to the reader which pig did not receive recumbency after 35 min). also the sentence line 222 is misleading, as it gives the impression, that all pigs became recumbent within 18.19 minutes. Please rewrite this paragraph

Paragraph 3.2-3.5 should have a heading “sedation score” – as these are just the 4 different time points assessed for  sedation scoring as outlined in paragraph 2.3.2

Please clarify – how had recovery times been assessed if they never reached lateral recumbency? As it is unclear, which groups/pigs did not achieved lateral recumbency, it is also difficult to take this into account when assessing recovery times. As these pigs were never recumbent, the time assessed for recovery is invalid/should not be used in this context. They might have been able to stand before carrying out the procedure, so how was that different from the condition after the procedure?

Do not repeat your results in the discussion. You also describe results you have never mentioned before (eg line 358)

Discussion could also be more to the point, and stick to relevant finding (eg flumazenil administration is not part of the study. Unclear why mentioned). Was there a difference in age between the groups ? (as you highlight that difference in your discussion, but according to line 218 – there is none) – relevance?

Rephrase conclusion – move the first sentence further to the end, if really needed. Your conclusion, is that of the various drug combinations only 1 protocol (G6) fit all criteria (
< 10 min induction time in average, < 60 min recovery time in average, < 1 in total sedation score) .there is nothing you can really highlight about all used intranasal protocols, as some of the IM protocols did not produce good results, either.

Comments on the Quality of English Language

komma instead of dot in some decimals
